# Oncologic and Long-Term Outcomes of Laparoscopic and Open Extended Cholecystectomy for Gallbladder Cancer

**DOI:** 10.3390/jcm11082132

**Published:** 2022-04-11

**Authors:** Jong Woo Lee, Jae Hyun Kwon, Jung Woo Lee

**Affiliations:** Department of Surgery, Hallym University Sacred Heart Hospital, Anyang 14068, Korea; jongw.lee0212@gmail.com (J.W.L.); ponakwon@gmail.com (J.H.K.)

**Keywords:** extended cholecystectomy, open surgery, gallbladder cancer, laparoscopic resection

## Abstract

Laparoscopic surgery has been traditionally contraindicated for gallbladder cancer, but there have been few reports demonstrating the oncologic outcomes of this treatment. This study aimed to compare the technical feasibility and the long-term outcomes after laparoscopic versus open extended cholecystectomy for gallbladder cancer. Between January 2011 and December 2018, 44 patients with gallbladder cancer who underwent extended cholecystectomy were included in this study, with 20 patients in the laparoscopic group and 24 patients in the open group. Perioperative outcomes, overall survival (OS), and recurrence-free survival (RFS) were retrospectively analyzed. There were no significant differences (*p* > 0.05) between the two groups in terms of perioperative outcomes, including blood loss, postoperative complications, R0 resection, and the number of lymph nodes retrieved. Patients in the laparoscopic group showed similar OS compared to the open group (5 year tumor-specific OS rate: 84.7% vs. 62.5%; *p* = 0.125). On subgroup analysis of patients with stage T2 and N0 disease, the laparoscopic group showed better OS (T2: 90.9% vs. 75.0%, *p* = 0.256; N0: 100.0% vs. 76.5%, *p* = 0.028). There was no difference in terms of RFS (3 year RFS: 74.4% vs. 64%; *p* = 0.571) and locoregional recurrence (10.0% vs. 16.9%, *p* = 0.895) between the two groups. There was no port-site recurrence in the laparoscopic group. This study suggests that laparoscopic extended cholecystectomy might be not inferior to open surgery in terms of oncologic safety or early and long-term outcomes in patients with early gallbladder cancer.

## 1. Introduction

Curative resection for gallbladder cancer (GBC) requires extended cholecystectomy, including liver resection and regional lymph node dissection. Owing to the difficulty of radical cholecystectomy and the concerns of tumor cell dissemination, the application of laparoscopy has been contraindicated in GBC. Laparoscopy is widely accepted in other digestive surgery since large controlled studies have shown that laparoscopic surgery provides significant benefit and does not increase port-site recurrence in stomach and colon cancer [1,2]. The reason for this contraindication was based on initial reports showing port-site and peritoneal recurrence after laparoscopic cholecystectomy for the treatment of GBC [3,4]. Recently, laparoscopic extended cholecystectomy has been increasingly performed, since experienced surgeons have reported favorable outcomes of laparoscopic surgery for GBC [5,6,7,8,9,10,11,12,13]. With improvements in laparoscopic techniques and instruments, the application of laparoscopic surgery for GBC has recently been extended to the advanced T3 stage [14].

GBC is increasingly diagnosed incidentally after cholecystectomy. Oncological extended resection is recommended for patients with T1b (or higher T stage) incidental GBC. Several recent studies [15,16,17] have reported that an initial laparoscopic approach did not decrease long-term survival in incidental patients with GBC if obvious R0 resection was achieved in the final pathology.

However, few studies have evaluated the long-term outcomes of laparoscopic surgery for GBC, and most of the previous studies are limited to case reports or small case series regarding technical feasibility, safety, and oncological outcomes. Therefore, the present study aimed to compare the technical feasibility and long-term outcomes (overall survival (OS) and recurrence-free survival (RFS)) of both procedures (laparoscopic vs. open extended cholecystectomy) in patients with GBC.

## 2. Materials and Methods

### 2.1. Patients

A total of 72 patients who underwent surgery for GBC at our hospital from January 2011 to December 2018 were identified and their electronic medical records were retrospectively reviewed. Of these patients, 18 who underwent simple cholecystectomy, 9 who underwent major hepatectomy or pylorus-preserving pancreaticoduodenectomy, and 1 who had concomitant pancreatic cancer were excluded. Finally, 44 patients who underwent extended cholecystectomy (20 and 24 patients in the laparoscopic and open groups, respectively) were included in this study.

Laparoscopic extended cholecystectomy for GBC was initially performed in January 2014 at our institution. Patients with suspected GBC of T2 (or below T stage) without evidence of liver, extrahepatic biliary tract, or adjacent organ invasion on preoperative imaging and without a history of major abdominal surgery were considered for laparoscopic surgery. If any of the above conditions were not met, laparotomy was performed.

Patients with incidentally proven GBC on postoperative pathology were also included in this study. Re-resection for incidental GBC was performed in all patients with stage T1b or greater depth of invasion. The laparoscopic approach for re-resection in patients with incidental GBC was first performed in 2015.

Tumor stage was classified according to the eighth edition of the American Joint Committee on Cancer (AJCC) staging manual [18]. For postoperative surveillance, abdominal computed tomography and laboratory tests, including tumor markers, were performed every 3 months for the first 2 years and every 6 months for the next 3 years. Patterns of recurrence were classified as locoregional (regional lymph node and liver cut surface) and distant (liver, peritoneum, and distant organs) metastases.

Based on the feasibility of both procedures, the number of samples size was calculated. The feasibility of laparoscopic and open extended cholecystectomy in early gallbladder cancer was set to 99%. When calculating the number of samples by setting alpha error to 0.025, power to 80% and non-inferiority margin (which means conversion rates as a threshold of laparoscopic feasibility) to 10% as the conventional level, at least 16 cases from each group were required, and in total 32 cases were required.

All procedures in this study were performed in accordance with the ethical standards of the institutional and/or national research committee, the Code of Ethics of the World Medical Association, and with the 1964 Declaration of Helsinki and its later amendments or comparable ethical standards. The study design was approved by the Institutional Review Board of our hospital. Written informed consent was obtained from all patients prior to each operative procedure.

### 2.2. Operative Procedures

For laparoscopic surgery, intraoperative ultrasonography was performed to detect liver invasion and occult liver metastasis. If the results suggested a T3 or higher stage of cancer, an open procedure was performed. In both approaches, first, dissection around the calot triangle was performed to expose the cystic artery and duct. The cystic duct was divided, and the cystic duct margin was sent to the Pathology Department for frozen-section analysis. If the margin was positive for malignancy, resection of the extrahepatic bile duct was performed. Limited resection of the liver bed or anatomical resection of the liver segments IVb and V was performed. After hepatectomy, regional lymph node dissection was performed, including dissection of the hepatoduodenal ligament, common hepatic artery, and peripancreatic head lymph nodes, as a standard approach for GBC.

In incidental GBC, oncological extended resection (re-resection) was performed in all patients with tumors of T1b stage or higher. Re-resection included exploration of the abdominal cavity, limited resection of the liver, and dissection of regional lymph nodes, similar to standard GBC.

### 2.3. Statistical Analyses

Categorical variables are expressed as frequency (percentage) and compared using the chi-squared (χ^2^) test or Fisher’s exact test. Continuous variables are presented as mean ± standard deviation and compared using the Wilcoxon rank-sum test. The Kaplan–Meier method and log-rank test were used to determine between-group differences in OS and RFS. Univariate and multivariate proportional hazards regression analysis was performed to examine the impact of prognostic factors on OS. All statistical analyses were performed using SPSS Statistics for Windows version 25.0 (IBM Inc., Armonk, NY, USA), with statistical significance set at *p* < 0.05.

## 3. Results

### 3.1. Patient Demographics and Baseline Clinical Data

This study analyzed 44 patients who underwent laparoscopic (*n* = 20) and open (*n* = 24) extended cholecystectomies. Patient characteristics are summarized in Table 1. There were no differences in most characteristics between the two groups, except for postoperative carbohydrate antigen 19-9 (CA19-9); CA19-9 levels were significantly lower in the laparoscopic group than in the open group (10.64 ± 7.40 vs. 20.68 ± 19.99, *p* = 0.04). Two patients who underwent laparotomy due to common bile duct invasion were included in the open group. Segment IVb and V resection was more frequently performed in the laparoscopic group, and limited resection of the gallbladder bed was more frequently performed in the open group; the liver resection type showed a significant difference between the two groups (*p* = 0.034). The proportion of patients with incidentally proven GBC was 35.0% (*n* = 7) in the laparoscopic group and 29.2% (*n* = 7) in the open group. There was no significant difference in median OS (51.28 ± 25.81 vs. 51.65 ± 34.29 months, *p* = 0.969) and median RFS (48.9 ± 27.76 vs. 46.46 ± 35.54 months, *p* = 0.804) between the two groups.

### 3.2. Perioperative Outcomes

Table 2 shows operative outcomes and no significant differences in operative time, blood loss, transfusion, postoperative complications, duration of hospitalization, or mortality between the laparoscopic and open groups. There were two cases (10.0%) of postoperative complications in the laparoscopic group, including bile leakage from the liver cut surface and complicated abdominal fluid collection in one patient each. There were five cases (20.8%) of postoperative complications in the open group, including abdominal fluid collection in four patients and pneumonia in one patient; however, the difference between the groups was not significant (*p* = 0.428).

### 3.3. Histopathologic Data and Tumor Stage 

Histopathological data are presented in Table 3. No significant differences were observed in most of the investigated histopathologic factors between the two groups. With respect to the Tumor-Node-Metastasis stage, T2 (60.0% vs. 66.7%), N0 category (70.0% vs. 70.8%), and stage II (55.0% vs. 50.0%) were the most frequent in both groups. Six patients (20% (*n* = 4) vs. 8.3% (*n* = 2)) were classified as Nx because the number of lymph nodes retrieved was <6 on pathologic examination. The two groups did not differ in terms of the number of removed lymph nodes (4.75 ± 3.54 vs. 5.75 ± 4.17, *p* = 0.402).

### 3.4. Recurrence Pattern and Treatment after Recurrence

Table 4 presents the recurrence data. Among the 44 patients, 31 patients were alive without recurrence at the last follow-up, whereas 13 patients (5 and 8 in the laparoscopic and open groups, respectively) showed evidence of recurrence on follow-up cross-sectional imaging. There were no port-site recurrences in the laparoscopic group. In both groups, the sites of recurrence were regional lymph nodes (10.0% vs. 12.5%), liver (10.0% vs. 8.3%), liver cut surface (0.0% vs. 8.3%), and peritoneum (10.0% vs. 8.3%). Furthermore, 40.0% and 100.0% of patients showing evidence of recurrence in the laparoscopy and open groups, respectively, received chemotherapy (*p* = 0.035).

### 3.5. Long-Term Outcomes

The OS (Figure 1) rates of the laparoscopic and open groups were compared in the overall cohort (*n* = 44) and in patients with stage T2 (*n* = 28) and N0 (*n* = 31) tumors. OS rates were significantly higher in the laparoscopic group than in the open group (5 year OS rate: 80% vs. 53.5%; *p* = 0.049) (Figure 1a). Among patients with stage T2 tumors, the 3 year OS rates were 83.3% and 75.0% in the laparoscopic and open groups, respectively; the difference was not statistically significant (*p* = 0.2) (Figure 1b). Among patients with stage N0 tumors, the 3 year OS rates were 92.9% and 76.5% in the laparoscopic and open groups, respectively; the difference was statistically significant (*p* = 0.024) (Figure 1c).

To exclude non-cancer-related deaths, tumor-specific deaths of both groups were investigated and additional survival analyses were performed. When comparing tumor-related deaths, survival rates were not significantly different between the two groups (5 year tumor-specific OS rate: 84.7% vs. 62.5%; *p* = 0.125) (Figure 2a). Among patients with stage T2 tumors, the 3 year OS rates were 90.9% and 75.0% in the laparoscopic and open groups, respectively; the difference was not statistically significant (*p* = 0.256) (Figure 2b). Among patients with stage N0 tumors, the 3 year tumor-specific OS rates were 100% and 76.5% in the laparoscopic and open groups, respectively; the difference was statistically significant (*p* = 0.028) (Figure 2c).

The RFS rates for both groups in the overall cohort (3 year RFS rate: 74.4% vs. 64.0%; *p* = 0.571), patients with stage T2 tumors (3 year RFS rate: 82.5% vs. 67.0%; *p* = 0.27), and patients with stage N0 tumors (3 year RFS rate: 92.3% vs. 75.3%; *p* = 0.39) showed no significant differences (Figure 3).

## 4. Discussion

Laparoscopic surgery has been considered a contraindication for GBC for a long time, largely because of technical difficulty; other factors include the effect that pneumoperitoneum could have on the spread of GBC and the higher risk of spread in case of bile leakage. However, it is widely accepted that the risk of tumor spread during surgery is low if precautions with gentle and careful manipulation are taken during the laparoscopic surgery [19,20].

In this study, anatomical resection of segment IVb and V was performed significantly more in the laparoscopic group (75% vs. 42%). There was no significant difference between the two groups in terms of cancer stage and tumor biologic factors except postoperative CA19-9 level. However, it seems that there are other possible factors that affected the selection of surgical method. First, there were more females with relatively small stature in the laparoscopic group, and conversely, the comorbidity of diabetes, hepatitis and liver cirrhosis was higher in the open group without statistical difference, which would have affected the decision for laparoscopic surgery. For this reason, more patients who were tolerable to anatomical liver resection may have been included in the laparoscopic group. In addition, segment IVb and V resection was preferred for achieving higher curability in this hospital. This institutional preference may also have affected preoperative planning.

The advantage of this study is that comparative assessment of early and long-term results is presented and subgroup analysis of patients with T2 and N0 disease. In terms of long-term results, OS was found to be significantly better in the laparoscopic group compared to the open group (5 year OS rate: 80% vs. 53.5%; *p* = 0.049). However, there was no difference between the two groups when comparing the tumor-specific OS in the overall cohort (5 year tumor-specific OS rate: 84.7% vs. 62.5%; *p* = 0.125). Considering all the results of this study, laparoscopic extended cholecystectomy was not inferior to open extended cholecystectomy in terms of early and long-term outcomes in early stages of GBC. 

To investigate prognostic factors affecting OS, univariate and multivariate analyses were performed (Appendix A). Age, male, liver resection type, intraoperative transfusion and T3 stage were shown to be significant prognostic factors in multivariate analysis. Although those factors were proven to be statistically significant, cautious interpretation of the results is important because of the small sample size and retrospective study nature.

Laparoscopic re-resection for incidental GBC is frequently performed in selected centers. However, there are still concerns that laparoscopic surgery could lead to the dissemination of tumor cells and incomplete resection of residual cancer [21]. This study included 14 patients (7 in the laparoscopic group and 7 in the open group) with incidental GBC (stage T1b or higher) diagnosed after laparoscopic cholecystectomy who underwent re-resection. There was no evidence of recurrence at the port-site or incisional site. The incidence of port-site metastasis in laparoscopic surgery is 10.3% in the modern era, which has decreased compared to an earlier study [4]. Maker et al. [19] reported that port-site resection did not improve OS or RFS and suggested that port-site metastasis was not related to the operative technique, but was rather a manifestation of disseminated disease. Based on these results, port-site resection was not routinely performed in patients with incidental GBC at our institution. Instead, surgeons strictly followed the principle of “no touch” for tumors, plastic bags were used for specimen retrieval, and massive saline irrigation was performed after resection to avoid tumor cells spreading outside the gallbladder.

Achieving a complete resection margin and sufficient lymph node retrieval are essential for evaluating oncologic outcomes. In this study, there was no significant difference in the positive rate of resection margins (5.0% vs. 4.2%; *p* = 1.000) and the number of lymph nodes retrieved (5.75 ± 3.54 vs. 6.75 ± 4.17; *p* = 0.402). The number of lymph nodes retrieved was <6 in six patients. It is recommended that at least six lymph nodes should be retrieved for accurate staging of GBC [18]. Although the mean number of lymph nodes retrieved in the laparoscopic group was slightly lower than six, this could be because patients who underwent surgery before publication of the eighth edition of the AJCC staging system were included in this study; the previous edition of the AJCC system recommended the removal of a minimum of three lymph nodes [22]. Therefore, our results show that laparoscopic surgery for GBC is feasible in terms of achieving adequate oncologic clearance.

The conversion rate from laparoscopic to open extended cholecystectomy was 9.1% in this study, suggesting acceptable laparoscopic skills. More than half of patients were converted to open surgery in an initial study of laparoscopic surgery for GBC for complete lymph node dissection [6]. In addition, in the present study, the morbidity (10.0%) and mortality (0.0%) rates in the laparoscopic group seemed to be acceptable and not inferior to those in the open group. With experience in laparoscopic liver and laparoscopic pancreatic surgery, laparoscopic resection of the bile duct has been recently performed, even when frozen-section examination of the cystic duct margin shows tumor cells. The progression of laparoscopic techniques has made it possible to adopt laparoscopic surgery for more advanced disease.

Several other studies [9,11,12,13,23] have reported the long-term outcomes in patients who underwent laparoscopic extended cholecystectomy for GBC. Most of these reports included fewer than 35 patients who underwent laparoscopic extended cholecystectomy. The survival data of each study was variable (5 year survival rate: 68.7–95.8%), which may be attributable to the heterogeneity of the patients or institutions. In this study, patients with stage T1 and T2 showed a 5 year survival rate of 93.0%, and patients with stage N0 showed a 5 year survival rate of 100.0%. These long-term results appear to be acceptable based on previous reports and suggest that laparoscopic surgery is not inferior to open surgery in terms of long-term oncologic outcomes in patients with GBC.

Patients undergoing laparoscopic surgery for GBC showed advantages in terms of postoperative pain and duration of hospital stay, although the differences were not significant. In addition, early recovery after laparoscopic surgery could enable early initiation of adjuvant therapy, which may affect survival. By reducing the use of postoperative analgesia, laparoscopic surgery could effectively reduce stress response, facilitate early ambulation, and restore gastrointestinal peristalsis, thereby shortening the length of hospital stay [24].

Although laparoscopic surgery should be considered for early-stage patients, open procedures may be more suitable for patients with advanced GBC. Therefore, preoperative patient selection is important. A previous study [25] suggested that high-risk prognostic features, including T3 disease, gallbladder perforation at the time of cholecystectomy, positive liver margins, and elevated CA19-9 levels, should be reviewed for safe selection for laparoscopic re-resection. Further research is needed to ensure appropriate patient selection for laparoscopic surgery for GBC.

This study had some critical limitations. First, patients were not randomized, and the retrospective nature of the study may have led to potential bias in the selection of patients and even the surgical method. Second, the number of patients in the overall cohort, and each group, was too small. Although laparoscopic extended cholecystectomy for GBC has been performed since 2014 in our institution, the total number of patients who underwent this type of surgery was still not large enough for detailed analysis. Therefore, it is difficult to draw definite conclusions because of low statistical power. Third, the follow-up period was not sufficiently long for the accurate evaluation of long-term survival rates. As the patient enrollment period was from January 2011 to December 2018, the follow-up period was approximately 3 years for some patients in this study. Future larger, prospective, and multicenter study is required because of these limitations.

## 5. Conclusions

The results of this study suggest that laparoscopic extended cholecystectomy for GBC might be not inferior to open surgery in terms of technical feasibility, oncologic safety, and long-term outcomes in early stages of GBC. Larger, randomized controlled trials could be helpful to obtain a clearer conclusion for minimally invasive surgery for GBC.

## Figures and Tables

**Figure 1 jcm-11-02132-f001:**
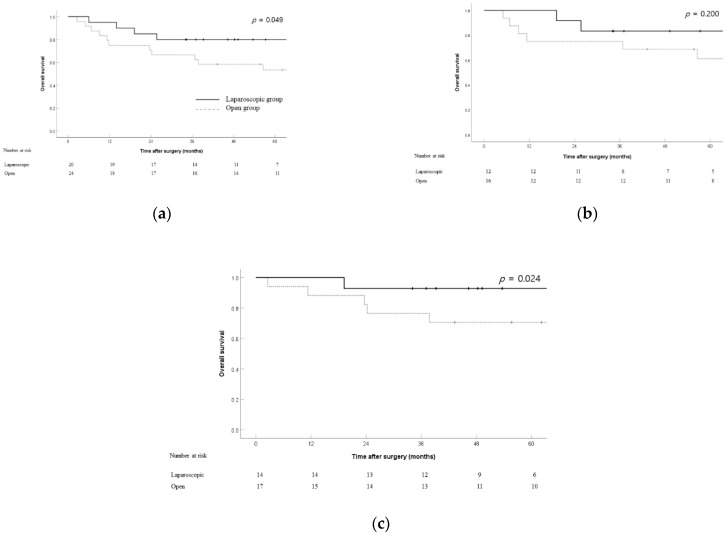
Overall survival between the laparoscopic and open groups in the overall cohort (*n* = 44) (**a**), T2 patients (*n* = 28) (**b**), and N0 patients (*n* = 31) (**c**).

**Figure 2 jcm-11-02132-f002:**
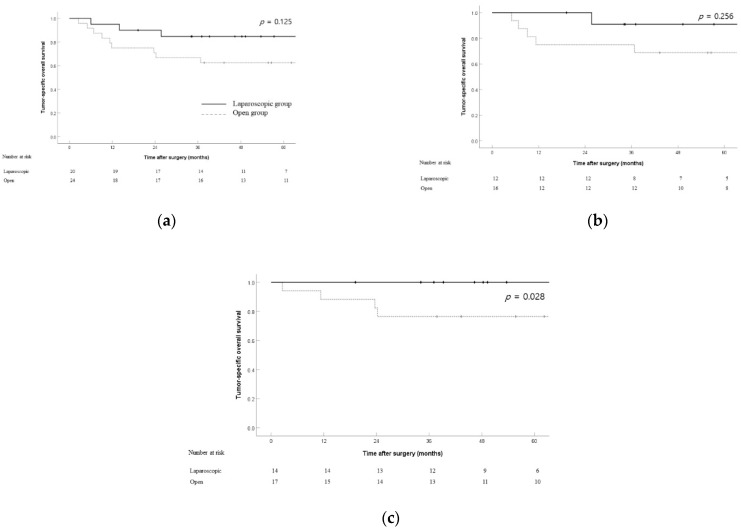
Tumor-specific overall survival between the laparoscopic and open groups in the overall cohort (*n* = 44) (**a**), T2 patients (*n* = 28) (**b**), and N0 patients (*n* = 31) (**c**).

**Figure 3 jcm-11-02132-f003:**
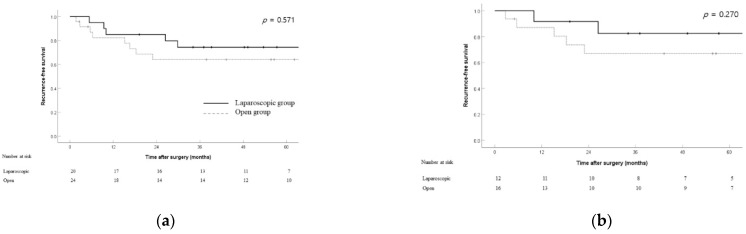
Recurrence-free survival between the laparoscopic and open groups in the overall cohort (*n* = 44) (**a**), T2 patients (*n* = 28) (**b**), and N0 patients (*n* = 31) (**c**).

**Table 1 jcm-11-02132-t001:** Baseline characteristics of patients with gallbladder cancer who underwent laparoscopic and open extended cholecystectomy.

	Laparoscopic Group (*n* = 20)	Open Group (*n* = 24)	*p*-Value
Age, years	71.85 ± 9.11	68.08 ± 10.64	0.219
BMI, kg/m^2^	24.88 ± 2.98	24.39 ± 2.70	0.568
Sex, *n* (%)			0.153
Male	5 (25.0)	11 (45.8)	
Female	15 (75.0)	13 (54.2)	
Diabetes mellitus, *n* (%)	7 (35.0)	10 (41.7)	0.651
Hypertension, *n* (%)	10 (50.0)	14 (58.3)	0.580
Hepatitis, *n* (%)	0 (0.0)	2 (8.3)	0.493
Liver cirrhosis, *n* (%)	0 (0.0)	1 (4.2)	1.000
Preoperative bilirubin, mg/dL	0.58 ± 0.26	0.83 ± 0.68	0.119
Preoperative albumin, g/dL	3.97 ± 0.46	3.90 ± 0.42	0.557
Preoperative CA19-9, U/mL	10.65 ± 7.51	24.56 ± 29.54	0.097
Postoperative CA19-9, U/mL	10.64 ± 7.40	20.68 ± 19.99	0.040 *
Incidental GBC, *n* (%)	7 (35.0)	7 (29.2)	0.679
Open conversion, *n* (%)	–	2 (8.3)	–
Adjuvant chemotherapy, *n* (%)	6 (30.0)	10 (41.7)	0.423
Liver resection, *n* (%)			0.034
No	1 (5.0)	1 (4.2)	
Wedge	4 (20.0)	13 (54.2)	
S4b/5	15 (75.0)	10 (41.7)	
Extrahepatic common bile duct resection, *n* (%)	1 (5.0)	3 (12.5)	0.614
Overall survival, months	51.28 ± 25.81	51.65 ± 34.29	0.969
Recurrence-free survival, months	48.9 ± 27.76	46.46 ± 35.54	0.804

* Statistical significant *p*-value (<0.05), BMI, body mass index; GBC, gallbladder cancer; CA19-9, carbohydrate antigen 19-9.

**Table 2 jcm-11-02132-t002:** Comparison of operative outcomes in laparoscopic and open group.

	Laparoscopic Group (*n* = 20)	Open Group (*n* = 24)	*p*-Value
Operative time, min	186.60 ± 88.14	231.67 ± 82.97	0.088
Estimated blood loss, mL	320.00 ± 451.72	593.75 ± 912.04	0.205
Transfusion, *n* (%)	6 (30.0)	12 (50.0)	0.179
R0 resection, *n* (%)	19 (95.0)	23 (95.8)	1.000
Postoperative complications, *n* (%)	2 (10.0)	5 (20.8)	0.428
Bile leakage	1 (5.0)	0 (0.0)	0.455
Abdominal fluid collection	1 (5.0)	4 (16.7)	0.673
Pulmonary complication	0 (0.0)	1 (4.2)	1.000
Postoperative hospital stay, days	10.95 ± 4.82	12.80 ± 4.87	0.216
90-day mortality, *n* (%)	0 (0.0)	1 (4.2)	1.000

**Table 3 jcm-11-02132-t003:** Comparison of histopathologic outcomes in laparoscopic and open group.

	Laparoscopic Group (*n* = 20)	Open Group (*n* = 24)	*p*-Value
Tumor size, cm	3.21 ± 1.90	3.64 ± 2.31	0.223
Perineural invasion, *n* (%)	3 (17.6)	2 (15.4)	1.000
Lymphovascular invasion, *n* (%)	3 (16.7)	2 (8.7)	0.638
Grade, *n* (%)			0.549
Well/moderate	15 (75.0)	14 (58.3)	
Poor/undifferentiated	3 (15.0)	5 (20.8)	
Unknown	2 (10.0)	5 (20.8)	
pT stage, *n* (%)			0.906
T1	4 (20.0)	5 (20.8)	
T2	12 (60.0)	16 (66.7)	
T3	4 (20.0)	3 (12.5)	
pN stage			0.457
N0	14 (70.0)	17 (70.8)	
N1	2 (10.0)	5 (20.8)	
Nx	4 (20.0)	2 (8.3)	
Pathologic stage (*n* = 37), *n* (%)			0.780
I	4 (20.0)	4 (16.7)	
II	11 (55.0)	12 (50.0)	
IIIA	2 (10.0)	2 (8.3)	
IIIB	2 (10.0)	4 (16.7)	
IVA	1 (5.0)	2 (8.3)	
Hepatic invasion, *n* (%)	3 (15.0)	3 (12.5)	1.000
Lymph node metastasis, *n* (%)	2 (10.0)	5 (20.8)	0.328
Number of retrieved lymph nodes	5.75 ± 3.54	6.75 ± 4.17	0.402

**Table 4 jcm-11-02132-t004:** Details of disease recurrence and treatment.

	Laparoscopic Group (*n* = 20)	Open Group (*n* = 24)	*p*-Value
Recurrence, *n* (%)			0.895
None	15 (75.0)	16 (66.7)	
Locoregional	2 (10.0)	4 (16.7)	
Distant	3 (15.0)	4 (16.7)	
Recurrence site, *n* (%)			
Port-site or incisional site	0 (0.0)	0 (0.0)	–
Regional lymph nodes	2 (10.0)	3 (12.5)	1.000
Liver	2 (10.0)	2 (8.3)	1.000
Liver cut surface	0 (0.0)	2 (8.3)	0.493
Peritoneum	2 (10.0)	2 (8.3)	1.000
Chemotherapy after recurrence (*n* = 13)	2/5 (40)	8/8 (100)	0.035 *

* Statistical significant *p*-value (<0.05).

## Data Availability

Not applicable.

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
