# Peer review of "Oncologic and Long-Term Outcomes of Laparoscopic and Open Extended Cholecystectomy for Gallbladder Cancer"

_jcm, 2022, doi:10.3390/jcm11082132_

Round 1
Reviewer 1 Report
The article compares laparoscopic and open extended cholecystectomy for gallbladder cancer. The topic remains open for discussion in the literature, since there are no convincing data on the oncological viability of laparoscopic access in this disease. A comparative assessment of early and long-term results is presented. The advantages of the study include a subgroup comparative analysis of survival in patients with T2N0M0 disease stage.
However, there are several remarks:
- Patients with suspected early-stage GBC with no evidence of liver, 62 extrahepatic biliary tract, or adjacent organ invasion were selected for laparoscopic approach, however, anatomical resections were performed more often in this group compared to the open group, and therefore the reason for this discrepancy should be specified.
- Overall survival was better in patients undergoing a laparoscopic procedure, while in a subgroup analysis of patients with stage T2NoMo, the authors found no difference in overall survival. This fact suggests a lower proportion of patients with more advanced tumor (greater than T2NoMo)/more aggressive disease in the laparoscopic group. However, there are no differences in TNM stage and other oncological factors (perivascular invasion, tumor differentiation, etc.) between the groups, which does not shed light on the differences in survival. The revealed significant differences in the level of the CA 19-9 oncomarker can hardly fully explain the situation, since the average values ​​in both groups were within the normal range.
- The authors point to one of the main shortcomings of the study: a small number of observations in both groups. This is indeed a serious shortcoming of the study, since, in addition to the small statistical power of the study, the small number of observations prevents a reliable regression analysis in order to assess the influence of factors on survival and exclude the heterogeneity of the compared groups.
- In the discussion, the authors mention regression analysis, the data of which are not presented in the "Results" section. It is not correct to discuss the results that are not presented in the corresponding section
Author Response
1. Patients with suspected early-stage GBC with no evidence of liver, 62 extrahepatic biliary tract, or adjacent organ invasion were selected for laparoscopic approach, however, anatomical resections were performed more often in this group compared to the open group, and therefore the reason for this discrepancy should be specified.
-> In this study, anatomical resection of segment IVb and V was significantly more performed in the laparoscopic group(75% vs 42%). There was no significant difference between the two groups in terms of cancer stage and tumor biologic factor (e.g.tumor size, lymphovascular invasion, differentiation) except postoperative CA19-9. However, it seems that there are other possible factors that affected to the selection of surgical method. First, there were more females with relatively small stature in the laparoscopic group, and conversely, the comorbidity of diabetes, hepatitis and liver cirrhosis was higher in the open group, which would have affected the decision for laparoscopic surgery. For this reason, more patients who were tolerable to anatomical liver resection may have been included in the laparoscopic group. In addition, segment IVb and V resection was preferred for achieving higher curability in this institution. This preference may also have affected in preoperative planning.
2. Overall survival was better in patients undergoing a laparoscopic procedure, while in a subgroup analysis of patients with stage T2NoMo, the authors found no difference in overall survival. This fact suggests a lower proportion of patients with more advanced tumor (greater than T2NoMo)/more aggressive disease in the laparoscopic group. However, there are no differences in TNM stage and other oncological factors (perivascular invasion, tumor differentiation, etc.) between the groups, which does not shed light on the differences in survival. The revealed significant differences in the level of the CA 19-9 oncomarker can hardly fully explain the situation, since the average values ​​in both groups were within the normal range.
-> In terms of long-term outcome, overall survival of overall cohort was significantly better in laparoscopic group compared to open group (Figure 1a). To exclude non-cancer-related deaths, tumor specific deaths of both group was investigated and additional survival analyses were performed. And there was no difference between the two groups when comparing the tumor-specific OS (Figure 2a).
3. The authors point to one of the main shortcomings of the study: a small number of observations in both groups. This is indeed a serious shortcoming of the study, since, in addition to the small statistical power of the study, the small number of observations prevents a reliable regression analysis in order to assess the influence of factors on survival and exclude the heterogeneity of the compared groups.
-> To investigate prognostic factors affecting on overall survival, univariate and multivariate analyses was performed and the results are attached to the supplementary table. Age, male, liver resection type, intraoperative transfusion and T3 stage were shown to be significant prognostic factor in multivariate analysis, but statistical power seems to be limited due to small sample size. In addition, cautious interpretation of the results is needed due to low statistical power.
4. In the discussion, the authors mention regression analysis, the data of which are not presented in the "Results" section. It is not correct to discuss the results that are not presented in the corresponding section
-> To investigate prognostic factors affecting on overall survival, univariate and multivariate analyses was performed and the results are attached to the supplementary table. I revised the sentences corresponding to the table
Reviewer 2 Report
Thank you for the possibility to review the article entitled “Oncologic and Long-Term Outcomes of Laparoscopic and Open Extended Cholecystectomy for Gallbladder Cancer”.
Overall, the article is very well written. I have no remarks on methodology, number of patients, and how results are presented. Also, the argument is interesting.
The main limitation is the retrospective design and the inherent selection bias, which is declared by the authors. Patients with negative prognostic factor were selected for open surgery and this could affect the results. However, perioperative outcomes and histopathological findings are well documented and were similar between groups.
A multivariate or a propensity score-matched analyses would be helpful to reduce section bias, however, as the cohort of patients is rather small, they are not feasible. Results should be taken and interpreted with care as they are presented.
As histopathological findings were similar, I do not truly understand where the significant OS difference between groups came from. I would add a third Kaplan-Meier analysis for tumor-specific overall survival, thus excluding non-cancer-related deaths. This would be very helpful in understanding the results.
Author Response
We thought that propensity score matching analysis was not appropriate because of small sample size. To investigate prognostic factors affecting on overall survival, univariate and multivariate analyses was performed and the results are attached to the supplementary table. Age, male, liver resection type, intraoperative transfusion and T3 stage were shown to be significant prognostic factor in multivariate analysis, but statistical power seems to be limited due to small sample size.
Overall survival of overall cohort was significantly better in laparoscopic group compared to open group (Figure 1a). As your comments, tumor specific deaths of both group was investigated and additional survival analyses were performed. The results were added as figure 2. There was no difference between the two groups when comparing the tumor-specific OS in overall cohort (Figure 2a).
Reviewer 3 Report
The aim of the study was to assess the impact of the surgical approach on long term outcomes in patients with gallbladder cancer.
For the purpose the Authors have reported a retrospective comparative study between two cohorts submitted to laparoscopic or open approach for GBC.
Comparative study is not the more appropriate study design to assess the impact on outcomes of a given procedure, an analytic study based on a univariate and multivariate analysis that takes into account confounding would be more appropriate.
However, based on the common rules of clinical research the use of laparoscopy for advanced GBC may be limited by a relevant ethical issue, if not done before, it must only be proposed within a regular registered clinical trial, such a reference has not been provided in the introduction. Indeed, the Authors compared laparoscopy for pre-operative early-stage disease with open surgery for cases lacking the criteria for early stage.
Moreover, the not randomized comparison, the high rate of excluded patients, differences in indications between the two groups, the high number of outcomes inducing revision of risk alfa threshold, are all limits that reduce the validity of the results. On a methodological point of view, according to the test of hypothesis rules, the lack of sample size calculation does not allow to consider laparoscopy comparable to open surgery. Without an evaluation of the risk beta, the absence of statistically significant difference does not allow to conclude in the absence of clinical difference, especially when comparison is performed between two limited cohorts as those reported by the Authors. The comparison should just be considered as non-conclusive.
Author Response
Comparative study is not the more appropriate study design to assess the impact on outcomes of a given procedure, an analytic study based on a univariate and multivariate analysis that takes into account confounding would be more appropriate.
-> To investigate prognostic factors affecting on overall survival, univariate and multivariate analyses was performed and the results are attached to the supplementary table. However, statistical power seems to be limited due to small sample size. In addition, cautious interpretation of the results is needed due to low statistical power.
However, based on the common rules of clinical research the use of laparoscopy for advanced GBC may be limited by a relevant ethical issue, if not done before, it must only be proposed within a regular registered clinical trial, such a reference has not been provided in the introduction. Indeed, the Authors compared laparoscopy for pre-operative early-stage disease with open surgery for cases lacking the criteria for early stage.
-> Laparoscopic surgery was considered for gallbladder cancer of stage T2 or below without evidence of liver, extrahepatic biliary tract, or adjacent organ invasion on preoperative image. The corresponding sentence was modified in the Materials and Method section.
Moreover, the not randomized comparison, the high rate of excluded patients, differences in indications between the two groups, the high number of outcomes inducing revision of risk alfa threshold, are all limits that reduce the validity of the results. On a methodological point of view, according to the test of hypothesis rules, the lack of sample size calculation does not allow to consider laparoscopy comparable to open surgery. Without an evaluation of the risk beta, the absence of statistically significant difference does not allow to conclude in the absence of clinical difference, especially when comparison is performed between two limited cohorts as those reported by the Authors. The comparison should just be considered as non-conclusive.
-> We mentioned that cautious interpretation of the results is important because of small sample size of this study, and conclusion was also slightly modified.
Based on the feasibility of both procedures, the number of samples was calculated. The feasibility of laparoscopic and open extended cholecystectomy in early gallbladder cancer was set to 99%, respectively. When calculating the number of samples by setting alpha error to 0.025, power to 80% and non-inferiority margin to 10% as the conventional level, at least 16 cases of each group was required, and a total of 32 cases were required.
Round 2
Reviewer 3 Report
Referring to the previous comments, comparisons between two non-randomized cohorts has been maintained, the suggested univariate and multivariate analysis has been reported as supplementary material but not within the manuscript. The multivariate analysis should consider one dependent variable, the desired outcome the Authors planned to explore and calculate its association with assessed risk factors according to a mathematical model that take into account assessed or well-known confusion factors. A so recommended analysis is not reported and explained within the method section for each explored outcome.
It is not clearly reported but it appears that between 2011 and 2014 all cholecystectomies were performed by open surgery, since 2014 pre-operative T2 stages have been approached by laparoscopy while more advanced cases by laparotomy. The mixed composition of the two cohorts do not allow to exclude important selection bias, moreover the unexplained exclusion of 18 simple cholecystectomies and 9 patients with unknown rate of major hepatectomies or pancreatectomies get selection bias worse.
Same comment for the sample size calculation: It is not reported within the section method and, as for multivariate analysis, it should be calculated for each outcome. The calculation based on laparoscopic and open feasibility is not clear, did the Authors consider a 10% conversion rate as a threshold of laparoscopic feasibility? Moreover, this sample size calculation does not appear to be correlated with the other reported outcomes within the conclusion, for which therefore, based on the common rule of the test hypothesis, the comparison should just be considered as non-conclusive.
Author Response
Referring to the previous comments, comparisons between two non-randomized cohorts has been maintained, the suggested univariate and multivariate analysis has been reported as supplementary material but not within the manuscript. The multivariate analysis should consider one dependent variable, the desired outcome the Authors planned to explore and calculate its association with assessed risk factors according to a mathematical model that take into account assessed or well-known confusion factors. Also recommended analysis is not reported and explained within the method section for each explored outcome.
->
As your suggestion, univariate and multivariate analysis for prognostic factor affecting overall survival was performed. In multivariate analysis, Age, male, liver resection type, intraoperative transfusion and T3 stage were shown to be significant prognostic factor. However, this table was left in the supplementary materials because I thought this analysis was less relevant to the main purpose of this study. The results and description of this table were mentioned in the Discussion section.
It is not clearly reported but it appears that between 2011 and 2014 all cholecystectomies were performed by open surgery, since 2014 pre-operative T2 stages have been approached by laparoscopy while more advanced cases by laparotomy. The mixed composition of the two cohorts do not allow to exclude important selection bias, moreover the unexplained exclusion of 18 simple cholecystectomies and 9 patients with unknown rate of major hepatectomies or pancreatectomies get selection bias worse.
->
The selection bias you mentioned is derived from the retrospective design of this study and this point is critical limitation of this study. I thought it was included in the sentence of the Discussion section, so I did not modify the manuscript. Future larger, prospective and multicenter study is required because of these limitations.
Same comment for the sample size calculation: It is not reported within the section method and, as for multivariate analysis, it should be calculated for each outcome. The calculation based on laparoscopic and open feasibility is not clear, did the Authors consider a 10% conversion rate as a threshold of laparoscopic feasibility? Moreover, this sample size calculation does not appear to be correlated with the other reported outcomes within the conclusion, for which therefore, based on the common rule of the test hypothesis, the comparison should just be considered as non-conclusive.
->
The open conversion rate in laparoscopic extended cholecystectomy was assumed to be 10%. Description of the sample size calculation was added to the Methods section.
As your comment that the comparison should be considered as non-conclusive, I also modified the conclusion to weaker sentence.
Thank you for your kind comments and suggestions.